behaviour/cognition

herring gulls, human–wildlife interactions, interspecific social learning, neophobia, object choice, social enhancement

**Author for correspondence:**
Madeleine Goumas
e-mail: m.goumas@exeter.ac.uk

# Urban herring gulls use human behavioural cues to locate food

Madeleine Goumas, Neeltje J. Boogert and Laura A. Kelley

Centre for Ecology and Conservation, University of Exeter, Penryn, Cornwall, UK

MG, 0000-0001-9338-2412; NJB, 0000-0002-1337-4365; LAK, 0000-0003-0700-1471

While many animals are negatively affected by urbanization, some species appear to thrive in urban environments. Herring gulls (*Larus argentatus*) are commonly found in urban areas and often scavenge food discarded by humans. Despite increasing interactions between humans and gulls, little is known about the cognitive underpinnings of urban gull behaviour and to what extent they use human behavioural cues when making foraging decisions. We investigated whether gulls are more attracted to anthropogenic items when they have been handled by a human. We first presented free-living gulls with two identical food objects, one of which was handled, and found that gulls preferentially pecked at the handled food object. We then tested whether gulls' attraction to human-handled objects generalizes to non-food items by presenting a new sample of gulls with two non-food objects, where, again, only one was handled. While similar numbers of gulls approached food and non-food objects in both experiments, they did not peck at handled non-food objects above chance levels. These results suggest that urban gulls generally show low levels of neophobia, but that they use human handling as a cue specifically in the context of food. These behaviours may contribute to gulls' successful exploitation of urban environments.

## 1. Introduction

Finding food is essential to survival but is potentially more challenging in changing environments. Humans have altered most environments extensively, and the ability of animals to adapt to human-mediated change may depend on behavioural traits that facilitate the use of anthropogenic resources [1], such as neophilia, boldness and the ability to learn quickly [2]. Whereas some species respond flexibly to endure increasing urbanization, others are less able to modify their behaviour to cope with the challenges that large-scale urbanization brings [3]. Urbanization can have a direct

effect on mortality rates; for instance, turtle hatchlings orient towards street lighting rather than the sea and are subsequently killed on roads [4], and various bird species are vulnerable to collisions with buildings [5].

Although living alongside humans affects many species negatively, it creates new opportunities for others. For example, house sparrows (*Passer domesticus*) and house martins (*Delichon urbica*) commonly nest in or on buildings [6,7], mammalian carnivores (Carnivora) across the world scavenge on human refuse [8] and geckos (Gekkota spp.) increase their feeding opportunities by exploiting the attraction of insects to artificial light [9].

Exploitation of anthropogenic resources in urban environments may be acquired through social learning, as in the case of blue tits (*Cyanistes caeruleus*) learning from each other how to peck through the foil caps on milk bottles to obtain cream [10,11]. Animals are expected to use social learning more than individual learning to locate food and other resources when their habitat is changing at a moderate pace [12], which might, for example, be caused by human activity and urbanization [13]. Additionally, an existing propensity to learn socially may enable animals to deal with challenges that are directly linked with human activity. For example, American crows can learn the facial features of dangerous humans and spread this information to naive conspecifics [14].

While social learning between conspecifics may help animals to thrive in urban environments, frequent interactions with humans could result in interspecific social learning from humans themselves. Such interspecific social learning could occur as a result of associations between human presence and strong reinforcers such as food [15]. It is possible that wild animals may learn to associate human behavioural cues, such as touching or gesturing, with the location of food. To date, research on the use of human behavioural cues by non-human animals has often focused on domesticated animals, and it has been suggested that domestication has selected for increased attentiveness towards humans [16]. Dogs, for example, can use a human's gaze direction to locate food [17], although their performance on this task has been mixed (e.g. [18]). Goats [19,20] and horses [21,22] can use human pointing cues to locate hidden food in object-choice tasks. Similar research has also been conducted on non-domesticated animals, with some primates (e.g. [23,24]), elephants [25,26], seals [27], dolphins [28], parrots [29] and corvids [30–32] using human cues to locate food. However, such research on human cue use has been limited to captive animals that often have extensive experience with human caretakers and trainers.

Wild animals that live alongside humans and make substantial use of anthropogenic resources are likely to have many opportunities to make use of human behavioural cues, but this has rarely been studied. Herring gulls are one such example of an animal species that has increased in numbers in urban areas and is often observed feeding on food discarded by humans [33]. We recently found that herring gulls are aware of human gaze direction when approaching a food source placed in close proximity to a human, and that they take longer to approach the food when human gaze is directed at them versus away [34]. However, it is not known whether gulls might actually learn from humans about novel foraging opportunities. In our previous study, gulls were attracted to the placement of a bag of chips on the ground [34]. It is possible that the gulls were attracted simply to the sight of food, but it is also possible that, through repeated exposure to humans, observing the act of the experimenter handling the food may have attracted their attention. Gulls may, therefore, use a form of social learning called 'local enhancement' [35] when foraging in areas populated by humans, whereby they are drawn to an object at a particular location after observing a human interacting with the object at that location.

We aimed to test whether human behavioural cues increase the probability of a gull interacting with an object, measured as the number of gulls making contact with the object by pecking at it. We first tested whether gulls would be more likely to peck at a food object that they had previously observed being handled by a human compared to an identical, non-handled food object. We then tested whether gulls would be attracted to any object previously handled by a human by repeating the experiment with similarly sized non-food objects. The aim of this second experiment was to determine whether human behavioural cues alone can attract gulls to peck at a particular item, or if herring gulls are only attracted by human behavioural cues when they are directed towards food objects.

## 2. Methods

### 2.1. Test subjects

We tested adult herring gulls in urban locations in South West England (approx. 50°N, 5°W; see electronic supplementary material, table S1 for location details). We selected individuals that were in

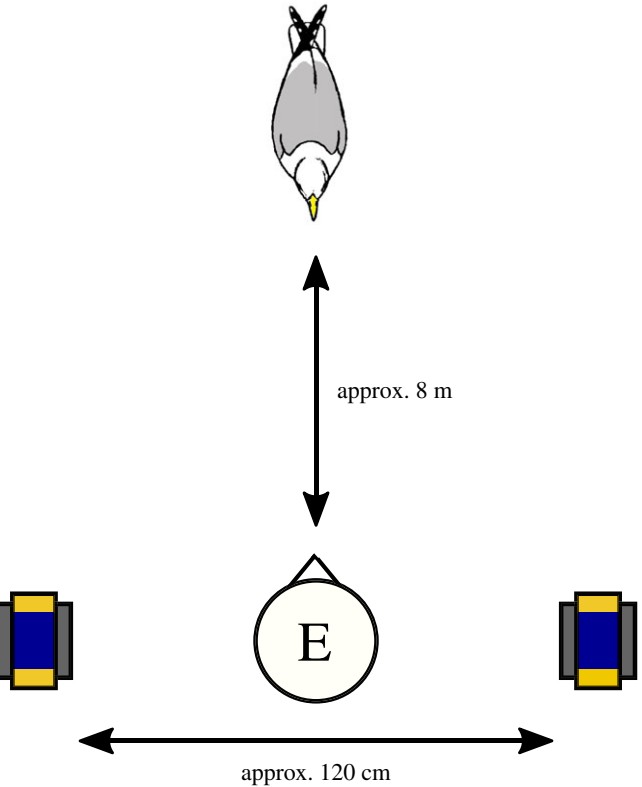

**Figure 1.** The experimental set-up. The experimenter (E) faced the gull and placed an upturned bucket, under which she held an object, either side of her body. She then removed the buckets to reveal the objects and picked up and handled one of the objects for 20 s before replacing it. Food objects (flapjacks in partially transparent, blue-coloured plastic wrappers, attached to grey slate tiles) depicted. Not to scale.

resting positions on the ground or on elevated structures (e.g. fences or lamp posts), and where the ground in front of the gull was flat and consisted of concrete, sand or short grass. Experiment 1 was conducted between 19th March and 28th May 2019, and Experiment 2 was conducted between 14th June and 25th July 2019. All trials were conducted by the same experimenter (MG) during daylight between 06.30 and 21.15 h, and were recorded by a second experimenter who used a Panasonic HC-V770 video camera mounted on a tripod and was positioned *ca* 10 m away from the objects presented in the experiments.

## 2.2. Experiment 1: Are gulls more attracted to the handled than non-handled food object?

The experimenter (MG) used two identical black plastic buckets (rim diameter 250 mm, 180 mm deep) to conceal two identical food items that were taped on top of, and weighed down by, dark grey slate tiles (100 × 100 mm, weight *ca* 250 g) to reduce the chance of the gulls flying off with them. The food items were 'Ma Baker' blueberry flapjacks (130 × 50 × 20 mm, 90 g) in their original plastic wrappers (electronic supplementary material, figure S3, left object in each picture). We chose these food items because they were identical in size, shape and appearance, with conspicuously coloured (blue) labels, and they were in transparent packaging that allowed the food to be partially seen.

MG held a bucket concealing the food/tile item (hereafter referred to collectively as a 'food object') in each hand and approached the gull so that it was approximately 8 m directly in front of her, at which point she placed the buckets on the ground with the food objects concealed underneath. To do this, she crouched down and outstretched her arms 90° to the left and right so that the food objects were equally spaced either side of her body and equidistant from the gull (figure 1). The food objects were positioned in the same orientation with the long axis of the flapjack pointing towards the gull and were not visible to the gulls before the removal of the buckets. MG wore dark glasses to avoid giving eye gaze cues.

After removing the two buckets to reveal both food objects simultaneously, MG placed the buckets behind her, picked up one of the two food objects and stood up. MG alternated handling the left

versus right object between completed trials. MG used a stopwatch to record a time of 20 s, during which she handled the object by picking it up and raising it up towards her face. She then repositioned the object in the same location, taking care to ensure that it remained in the same orientation as the other object, picked up the buckets and retreated to a position *ca* 10 m away. If gulls moved while MG was handling the object, she oriented her body so that she continued facing the gull and the two objects remained equidistant at right angles to her body upon replacement of the handled object. MG mentally noted the position of the gull at the time when she replaced the handled object, and this was verified with the video footage. She monitored the gull for an approach within 120 s of the object being repositioned and recorded which object the gull pecked at. The time taken for approaching gulls to peck at an object was recorded. A trial was considered 'complete' when a gull pecked at one of the presented objects. The experimenter terminated the trial if the gull walked or flew away.

Immediately after each trial, MG measured the distance between the presented food objects and the gull, and the gull's elevation from the ground, at the point in time when she replaced the handled object, in case these variables affected the gulls' choices through differences in viewing distance. Because there was also some variation in how far apart the objects were placed, MG also measured the distance between the objects after complete trials. We avoided conducting trials when there were humans or conspecifics other than mates (see the electronic supplementary materials) in close proximity (within *ca* 10 m from the objects or focal gull) and ended trials if the gulls were disturbed by humans or other animals. These trials were therefore not used in analyses. After each trial, the food objects were checked and replaced if damaged.

For incomplete trials in which gulls did not peck at either of the objects, we measured the distance between the objects and the gull at the time of the object being replaced and the elevation of the gull from the ground. These incomplete trials included gulls that remained in their original location for 120 s after the food object presentation and those that approached the objects by walking towards them but did not peck at either object. For all trials, we recorded the time of day to account for daily variation in motivation to feed and/or approach objects, and whether the gull's mate was present in case this affected the focal gull's behaviour. We also recorded the number of gulls that flew or ran away upon MG initiating the trials.

## 2.3. Experiment 2: Are gulls more attracted to the handled than non-handled non-food object?

Having found an effect of human handling on herring gulls' choice of food objects (see Results), we tested whether this effect would generalize to non-food objects. We used blue sponges cut to the same size and shape as the food objects (weight 10 g) and repeated the above experiment with a new sample of gulls so that subjects would not be familiar with the experimental set-up. By choosing different test locations, we could reliably ascertain that none of these gulls had been tested in the first experiment, owing to the territoriality of herring gulls [36].

## 2.4. Statistical analyses

Statistical analyses were conducted in R version 3.5.3 [37]. For each experiment, we used a generalized linear model with a binomial error distribution to test whether gulls' choice of object (left/right) was influenced by which object the experimenter had handled (left/right). The model included the following potential confounds: the distance between the two objects, the distance between the objects and the gull and the elevation of the gull from the ground.

To test which factors affected whether or not gulls approached the objects (regardless of whether the gulls pecked at them), we used a generalized linear model with binomial error distribution on the data for both experiments combined and included the following variables as predictors of approaching an object (yes/no): object type (food versus non-food), the distance between the objects and the gull, the elevation of the gull from the ground at the time of the object being replaced, the time of day and whether the gull's mate was present. As some gulls approached the objects without pecking at either of them, we used another generalized linear model with binomial error distribution to test whether the same variables affected whether or not approaching gulls pecked at either of the presented objects. We report here the results of the full models and the odds ratios (OR) of each predictor (the exponential of the regression coefficient). An OR of 1 indicates that exposure to an experimental treatment (e.g. handling) has no effect on the odds of an outcome of interest occurring. An OR > 1 indicates that the treatment is associated with a higher odds of the outcome occurring and an OR < 1 (bounded by 0) indicates that the treatment is associated with a lower odds of the outcome occurring.

**Table 1** . The results of Experiments 1 and 2 testing herring gulls' use of human handling as a cue when choosing between two identical objects. Section (*a*) shows the results of the food object trials, and (*b*) shows the results of the non-food object trials. The effects of potential confounds are also shown. Significant predictors are printed in italics.

|  | estimate | s.e. | Z | odds ratio | p |
|---|---|---|---|---|---|
| (*a*) food objects (*n* = 24) | | | | | |
| intercept | −1.263 | 6.978 | −0.181 | — | 0.856 |
| *handling* | *3.006* | *1.369* | *2.196* | *20.199* | *0.028* |
| distance between objects | −0.011 | 0.055 | −0.197 | 0.989 | 0.844 |
| distance to gull | 0.000 | 0.003 | 0.091 | 1.000 | 0.928 |
| starting height of gull | 0.001 | 0.004 | 0.275 | 1.001 | 0.783 |
| (*b*) non-food objects (*n* = 23) | | | | | |
| intercept | −4.303 | 7.607 | −0.566 | — | 0.572 |
| handling | 2.013 | 1.701 | 1.183 | 7.484 | 0.237 |
| distance between objects | 0.038 | 0.065 | 0.585 | 1.039 | 0.559 |
| distance to gull | 0.000 | 0.004 | 0.086 | 1.000 | 0.932 |
| starting height of gull | −0.004 | 0.005 | −0.764 | 0.996 | 0.445 |

To determine whether gulls' behaviour might be affected by their perception of the two different objects used in Experiments 1 and 2, we quantified their appearance in terms of visual contrast and visual acuity using avian visual models (see electronic supplementary material, methods).

# 3. Results

## 3.1. Experiment 1: Are gulls more attracted to the handled than non-handled food object?

We presented 38 herring gulls with the two food objects. Twenty-six gulls approached the objects and 24 pecked at one of the objects (see electronic supplementary material, table S1). Human handling of a food object had a significant effect on the gull's choice of which object to peck at (binomial GLM, OR = 20.199, $Z = 2.196$, $p = 0.028$; table 1): 19 (79%) of the 24 participating gulls pecked at the food object that the experimenter had handled. There was no significant effect on food object choice of the gull's distance from the objects, the gull's elevation from the ground or the distance between the objects (table 1).

## 3.2. Experiment 2: Are gulls more attracted to the handled than non-handled non-food object?

After completing the food object trials, we presented 41 experimentally naive herring gulls with the two non-food objects (blue sponges cut into the same size and shape as the flapjacks presented in Experiment 1). Thirty-two gulls approached the objects and 23 pecked at one of these objects (see electronic supplementary material, table S1). Fifteen (65%) of these gulls pecked at the handled non-food item, which was not significantly different from chance levels (binomial GLM, OR = 7.484, $Z = 1.183$, $p = 0.237$; table 1). There was no significant effect of the gull's distance from the objects, the gull's elevation from the ground or the distance between the objects (table 1).

## 3.3. Do gulls behave differently towards food versus non-food objects?

All gulls that pecked at one of the two presented objects did so within 42 s of the experimenter replacing the handled object. There was no significant difference in the time taken for a gull to peck at an object in each experiment (mean ± s.d., food objects: 18.5 ± 2.07 s, non-food objects: 17.9 ± 2.15 s; see electronic supplementary material, table S3).

There was no significant difference in the number of gulls that approached the non-food objects compared to the food objects (binomial GLM, OR = 0.830, $Z = −0.255$, $p = 0.799$; food object trials: 26 of 38 gulls, non-food object trials: 32 of 41 gulls). Whether gulls approached the objects was not significantly affected by the time of day (OR = 0.999, $Z = −0.937$, $p = 0.349$), the elevation of the gull

from the ground at the start of the trial (OR = 0.998, Z = −1.409, p = 0.159) or whether the mates of gulls were present during the trials (OR = 1.061, Z = 0.067, p = 0.946). However, the distance between the objects and the gull at the time the experimenter replaced the handled object was a significant predictor of whether gulls approached the objects (OR = 0.996, Z = −3.043, p = 0.002), with gulls significantly less likely to approach when objects were placed further away from them.

Of those gulls that did approach the objects, significantly fewer pecked at an object in the non-food trials than the food trials (binomial GLM, OR = 0.163, Z = −0.255, p = 0.046; food object trials: 24 of 26 gulls, non-food object trials: 23 of 32 gulls). There was no significant effect of the time of day (OR = 1.002, Z = 1.263, p = 0.207), the distance between the objects and the gull (OR = 0.997, Z = −1.343, p = 0.179), the elevation of the gull from the ground (OR = 1.004, Z = 1.075, p = 0.283) or whether the gull's mate was present (OR = 1.747, Z = 0.472, p = 0.637).

## 3.4. Perception of food and non-food objects

Both food and non-food items were easily discriminable from the grey tile background, and each other, in both colour and luminance (see electronic supplementary material, table S4). The non-food object was particularly salient against the grey background tile. Gulls could also visually resolve details of the food and non-food objects throughout the trials (i.e. at distances ranging from 30 cm to 8 m) based on our acuity analysis (electronic supplementary material, figure S3).

# 4. Discussion

Despite interactions between humans and wildlife becoming increasingly common, little research has been conducted on how wild animals may use direct human cues to exploit anthropogenic resources in urban environments. Here, we tested whether herring gulls use human behaviour to locate food. Gulls were significantly more likely to peck at a food object that a human had handled than an equally accessible, identical object that had not been handled. This shows that human handling of food attracts the attention of gulls and that handled food is more attractive than food that gulls have not observed being handled.

To determine whether this attractive effect of human handling was a result of the experimenter drawing attention to the presence of food or if handling alone was sufficient to motivate gulls to peck at the objects, we repeated the experiment using an identical protocol but instead presented non-food objects. Gulls did not peck at the handled non-food object above chance levels, suggesting that the appearance of food is likely to be particularly important in drawing gulls' attention to a specific object or location.

Although more gulls pecked at the handled food object compared to the handled non-food object, the total number of gulls pecking at either of the objects was similar in the food and non-food object trials. This indicates that, while visual cues of food appear to be important in making foraging decisions, gulls are also attracted to objects without these food cues. Food cues may include the appearance of the food itself as well as the plastic packaging that is used to wrap many different types of food items and thus may be associated with food.

Despite similar numbers of gulls approaching food and non-food objects, more gulls approached the objects without pecking at them in the non-food trials than in the food trials. This suggests that gulls may approach objects before distinguishing what they are, and discriminate between types of object at a closer distance. It is improbable that the gulls had previously encountered the exact food and non-food objects we presented, and it is likely that they were initially attracted to both types of object to determine whether they contained or were composed of food. Our results imply that, while gulls are attracted to non-food objects, and many peck at them, they may be more selective or cautious once they can observe such objects more closely.

It is perhaps not surprising that more gulls pecked at the objects associated with a food reward, but it is difficult to determine why so many gulls also pecked at the non-food objects. It may be worthwhile for urban herring gulls to peck at novel objects of any type if there is a chance that they could contain food. It is possible that the gulls that pecked at the non-food objects did so because these objects did not appear sufficiently different from food, but this seems unlikely because food is rarely the colour of the objects we chose (completely blue sponges), nor were the objects shiny as in the case of most food packaging. Furthermore, our visual models demonstrate that gulls could visually discriminate between the food and non-food items, but also that the non-food item was more salient against the visual background and so may have been more conspicuous or attractive.

There may be several reasons why some of the gulls pecked at objects that were not handled by the experimenter. Firstly, it is conceivable that the presence of the experimenter alone was sufficient to create an effect of local enhancement [35], with gulls' attention being drawn to the general location of the object presentation including the non-handled object as it remained in view while the experimenter handled the other object close by. In a study of horses, test subjects were more likely to choose feed buckets that were in close proximity to the experimenter [38], indicating that direct contact with the object is not necessary to generate an effect. There may also have been an effect of stimulus enhancement, whereby a demonstrator's interaction with an object results in an observer being more likely to interact with an object of the same type [35]: if gulls saw that the handled object was identical to the non-handled object, they may have been drawn to either object equally as there would be no apparent difference in consequence. Gulls may have pecked at handled food objects more often than handled non-food objects owing to having learned from previous experiences in their urban habitat that food packages are usually opened by humans and thus handled food objects tend to be more profitable. However, it is far less likely that the gulls would have had previous experiences of human handling making food accessible from the type of novel, non-food objects we presented.

As many gulls approached and pecked at novel objects, this implies that they have a low level of neophobia (fear of novelty), and could even be neophilic (attracted to novelty), which may facilitate their successful exploitation of urban environments [2]. High exposure to anthropogenic items could have influenced this behaviour, with gulls having perhaps learned that objects of a wide variety of shapes, sizes and colours may have food concealed inside, and tests specifically aimed at measuring neophobia or neophilia would be required to fully understand gulls' perception of novel objects [39]. It is possible that urban-living gulls may categorize anthropogenic items by similarities in physical features (e.g. size, shape, material) in the same way that jackdaws (*Corvus monedula*) in urban areas appear to be able to categorize food litter [40]. In addition, gulls may be attracted to handled food objects not because of the appearance of food but because food packaging is associated with food.

Our experimental set-up potentially selected the least fearful individuals in the population, as only those that did not flee when the experimenter approached or placed the buckets were tested. Consequently, the patterns in behaviour may not be representative of all urban-living herring gulls, and may not be representative of gulls living in areas that are less populated by humans. Nevertheless, the individuals we tested are the ones that are most inclined to interact with humans and potentially be involved in 'nuisance' behaviour [2]. Increasing our understanding of these individuals and how they make foraging decisions will be beneficial in generating methods to reduce negative interactions between humans and herring gulls without compromising this species' conservation status.

Other research on human cue use by wild animals has mainly been restricted to animals that have been to some degree, and sometimes extensively, socialized with humans [41]. It is unlikely that herring gulls would use human cues had they not had previous experience of humans and associated human activity with food litter. Although studies on social learning have largely focused on intraspecific information use, it is widely recognized that social learning between heterospecifics is widespread and confers many of the same benefits as intraspecific social learning, as well as other benefits such as reduced competition [42]. Gulls rely extensively on conspecifics to locate food and often procure food after watching other gulls flocking to food sources [43]. Using humans as an additional source of information is likely to be advantageous if there is a reliable association between humans and the availability of food.

The previous research on human cue use by other animals has usually involved object-choice tasks in which food is hidden and not directly touched by humans (e.g. [20,29]). The animals in these studies are usually tested on their understanding and use of gestures rather than direct experimenter handling of an object, and as such our results cannot be directly compared. However, our study is similar in design and results to that conducted by Schloegl *et al*. [30], where captive, hand-raised ravens preferentially touched objects that had been handled by the experimenter, and indicates that free-living animals are able to learn from humans in a similar manner to captive animals. Research that assesses the relative importance of human behavioural cue use to animals in urban environments and the ontogeny of such behaviour will aid in understanding the ecological drivers and cognitive mechanisms of learning from humans.

It is highly unlikely that herring gulls are the only wild animals to use human behavioural cues in urban areas. As urbanization increases, more wild animals will come into contact with humans and anthropogenic items. There may be an increased number of incidences of individuals of certain species displaying problematic behaviour, which can create conflicts between human activity and conservation [44]. Additionally, although purposeful provisioning of wildlife may in certain cases appear to be beneficial (such as the feeding of garden birds [45]), being attracted to anthropogenic items and feeding

on anthropogenic food can be harmful for wildlife (e.g. [46,47]). A more comprehensive understanding of the cues that cause wild animals to engage in interactions with humans is likely to be key in developing preventative measures that not only reduce negative encounters for humans but also potentially lessen the impact of anthropogenic items on wild animal populations.

Ethics. This work was approved by the University of Exeter Ethics Committee (ref.: eCORN000344).

Data accessibility. The data and R code used for the statistical analyses can be found at https://doi.org/10.5061/dryad.bnzs7h469.

Authors' contributions. M.G. conceived the idea and developed the protocol with N.J.B. and L.A.K. M.G. conducted the experiments and performed the statistical analyses. L.A.K. conducted the visual analyses. M.G. wrote the first draft and all authors contributed to the content of the final manuscript. All authors gave approval for publication and agreed to be held accountable for the content of the manuscript.

Competing interests. We declare that we have no competing interests.

Funding. N.J.B. and L.A.K. are funded by Royal Society Dorothy Hodgkin Research Fellowships. (N.J.B.: DH140080, L.A.K.: DH160082.)

Acknowledgements. We thank Tom Holding, Thomas Collins, Emma Inzani, Angharad Jeremiah, Leo Fordham and Drew Baigent for filming the trials, and Michael A. Patten and an anonymous reviewer for providing useful comments on the manuscript.

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
