## [Reviewer comments · Royal Society Open Science]

Review History

RSOS-191959.R0 (Original submission)

Review form: Reviewer 1 (Michael Patten)

Is the manuscript scientifically sound in its present form?

No

Are the interpretations and conclusions justified by the results?

Yes

Is the language acceptable?

Yes

Do you have any ethical concerns with this paper?

No

Have you any concerns about statistical analyses in this paper?

Yes

Recommendation?

Major revision is needed (please make suggestions in comments)

Comments to the Author(s)

Title: Urban herring gulls use human behavioural cues to locate food

Authors: Madeleine Goumas, Neeltje J. Boogert, and Laura A. Kelley

The authors established an experiment to assess whether Herring Gulls in an urban setting found items handled by humans to be more attractive (i.e., more worthy of immediate attention). Handled food was more attractive than untouched food. Attraction broke down, though, if one item was not food.

General comments:

The design of the first experiment is straightforward. I expect that the analysis is sufficient to support the reported effect of more attraction (or curiosity) toward food handled by humans than food not handled by humans. Some may argue that the statistic chosen is a bit convoluted, but it strikes me as acceptable given the need to consider covariates. My sole suggestion is that the authors not rely so heavily of the p-value. I cannot deny that it has long been acceptable to report a small p-value, reject H_0 , and conclude H_A is supported, but it remains that what matters is not the probability of a type I error so much as how biologically meaningful is the result. The authors therefore ought to report an estimate of the effect size. This estimate is not trivial for a logistic regression model (I use this term because a GLM with binomial errors and a logistic regression are interchangeable in terms of maximum likelihood estimators generated), yet neither are they particularly difficult to calculate (see, for example, Allen & Le 2008, *J. Edu. Behav. Stat.* 33:416–441). Short of that, an effect size estimate for a basic observed vs. expected goodness-of-fit test, with expected assumed to be equal pecks at handled or not, is easy to obtain. The need for estimates of effect size are underscored by the authors own remark (p. 11) that “Although more gulls pecked at the handled food object compared to the handled non-food object, the total number of gulls pecking at either of the objects was similar in the food and non-food object trials.” If the effect size was roughly the same, then conclusions ought to be revised.

I found results of the second experiment less compelling, perhaps because I wonder about the curiosity factor. I agree that a key difference is that the item presented in exp. 1 was food whereas in exp. 2 it was not, but another key difference, if I understand the experimental set up properly and judging from images in Fig. 2, is that the items in the first instances were wrapped whereas the item in the second instance was not. Would presentation of a wrapped non-food item have elicited comparable results to exp. 1? Similarly, would presentation of an unwrapped food item in exp. 1 have elicited different results? Could it be that urban gulls are naturally curious about a wrapped item – the can discern something is inside – and more likely to peck at a wrapped item another organism handled? In asking these questions I do not mean to suggest that results ought to be called into question but rather this possibility be discussed.

Lastly, and with all due respect for the hard work it entailed, I do not see how the visual analysis adds appreciably to the story. Put another way, if it were removed there would be little need to change much of the Results or Discussion.

–Michael A. Patten
Oklahoma Biological Survey
University of Oklahoma

Review form: Reviewer 2

Is the manuscript scientifically sound in its present form?

No

Are the interpretations and conclusions justified by the results?

No

Is the language acceptable?

Yes

Do you have any ethical concerns with this paper?

No

Have you any concerns about statistical analyses in this paper?

No

Recommendation?

Major revision is needed (please make suggestions in comments)

Comments to the Author(s)

The authors present the results of two studies in which an urban population of herring gulls are offered the choice between two 'food' items or two non-food items. In the first experiment, gulls were more likely to peck at a food item that had been handled by a stimulus human. In the second experiment, handling had no effect on the willingness of gulls to peck at a non-food item. While this is an interesting question, insufficient detail is given in the methods to adequately assess the merits of the study.

Additional details are needed regarding the focal subjects. What was the local density of gulls for each observation? Social information theory predicts differential use of public information based on costs. Competitive costs would likely be much higher (leading to a greater use of social information) in aggregations vs. solitary or small groups of birds. Likewise, social information use differs by sex, ontogenetic stage, general health, recent foraging success (and several other factors). Information regarding focal birds is needed to account for potential confounding factors.

L129-133: Some information on the number of trials that not included in the analysis is needed. Overall, how likely was it that gulls simply did not approach the food items, and was this confounded by the presence of competitors?

L121: how reliable is this 'mental notation'?

L134-141: the behavioural measures need to be properly (and completely defined). How long did it take for the gull to approach? Was this included as a dependent variable?

L155: as described in the methods, the objects were presented at a uniform distance. Why is this included as a factor in the model?

L 241: define JND

The discussion is often highly speculative. For example, what is the justification that the food items provided are novel (L276)? Presumably, gulls are pretty good at generalizing food preferences; how certain are you that they had not previously foraging on a similar food item? Likewise, lines 308-314 speculates on the role of individual differences... was this in anyway quantified? Generally, the discussion is overly long (5 pages) and could easily be shortened considerably to focus only on the relevant data.

Decision letter (RSOS-191959.R0)

17-Dec-2019

Dear Miss Goumas,

The editors assigned to your paper ("Urban herring gulls use human behavioural cues to locate food") have now received comments from reviewers. We would like you to revise your paper in accordance with the referee and Associate Editor suggestions which can be found below (not including confidential reports to the Editor). Please note this decision does not guarantee eventual acceptance.

Please submit a copy of your revised paper before 09-Jan-2020. Please note that the revision deadline will expire at 00.00am on this date. If we do not hear from you within this time then it will be assumed that the paper has been withdrawn. In exceptional circumstances, extensions may be possible if agreed with the Editorial Office in advance. We do not allow multiple rounds of revision so we urge you to make every effort to fully address all of the comments at this stage. If deemed necessary by the Editors, your manuscript will be sent back to one or more of the original reviewers for assessment. If the original reviewers are not available, we may invite new reviewers.

- Data accessibility

<http://datadryad.org/submit?journalID=RSOS&manu=RSOS-191959>

- Competing interests

- Authors' contributions

- Acknowledgements

- Funding statement

on behalf of Prof Kevin Padian (Subject Editor)
openscience@royalsociety.org

Associate Editor's comments:

The two reviewers have recommended a number of revisions - please ensure you thoroughly address these, as the journal does not routinely permit multiple rounds of revision.

Comments to Author:

Reviewers' Comments to Author:

Reviewer: 1

Comments to the Author(s)

Title: Urban herring gulls use human behavioural cues to locate food
Authors: Madeleine Goumas, Neeltje J. Boogert, and Laura A. Kelley

The authors established an experiment to assess whether Herring Gulls in an urban setting found items handled by humans to be more attractive (i.e., more worthy of immediate attention). Handled food was more attractive than untouched food. Attraction broke down, though, if one item was not food.

General comments:

The design of the first experiment is straightforward. I expect that the analysis is sufficient to support the reported effect of more attraction (or curiosity) toward food handled by humans than food not handled by humans. Some may argue that the statistic chosen is a bit convoluted, but it strikes me as acceptable given the need to consider covariates. My sole suggestion is that the authors not rely so heavily of the p-value. I cannot deny that it has long been acceptable to report a small p-value, reject H_0 , and conclude H_A is supported, but it remains that what matters is not the probability of a type I error so much as how biologically meaningful is the result. The authors therefore ought to report an estimate of the effect size. This estimate is not trivial for a logistic regression model (I use this term because a GLM with binomial errors and a logistic regression are interchangeable in terms of maximum likelihood estimators generated), yet neither are they particularly difficult to calculate (see, for example, Allen & Le 2008, *J. Edu. Behav. Stat.* 33:416–441). Short of that, an effect size estimate for a basic observed vs. expected goodness-of-fit test, with expected assumed to be equal pecks at handled or not, is easy to obtain. The need for estimates of effect size are underscored by the authors own remark (p. 11) that “Although more gulls pecked at the handled food object compared to the handled non-food object, the total number of gulls pecking at either of the objects was similar in the food and non-food object trials.” If the effect size was roughly the same, then conclusions ought to be revised.

I found results of the second experiment less compelling, perhaps because I wonder about the curiosity factor. I agree that a key difference is that the item presented in exp. 1 was food whereas in exp. 2 it was not, but another key difference, if I understand the experimental set up properly and judging from images in Fig. 2, is that the items in the first instances were wrapped whereas the item in the second instance was not. Would presentation of a wrapped non-food item have elicited comparable results to exp. 1? Similarly, would presentation of an unwrapped food item in exp. 1 have elicited different results? Could it be that urban gulls are naturally curious about a wrapped item – the can discern something is inside – and more likely to peck at a wrapped item another organism handled? In asking these questions I do not mean to suggest that results ought to be called into question but rather this possibility be discussed.

Lastly, and with all due respect for the hard work it entailed, I do not see how the visual analysis adds appreciably to the story. Put another way, if it were removed there would be little need to change much of the Results or Discussion.

–Michael A. Patten
Oklahoma Biological Survey
University of Oklahoma

Reviewer: 2

Comments to the Author(s)

The authors present the results of two studies in which an urban population of herring gulls are offered the choice between two ‘food’ items or two non-food items. In the first experiment, gulls were more likely to peck at a food item that had been handled by a stimulus human. In the second experiment, handling had no effect on the willingness of gulls to peck at a non-food item. While this is an interesting question, insufficient detail is given in the methods to adequately assess the merits of the study.

Additional details are needed regarding the focal subjects. What was the local density of gulls for each observation? Social information theory predicts differential use of public information based on costs. Competitive costs would likely be much higher (leading to a greater use of social information) in aggregations vs. solitary or small groups of birds. Likewise, social information use differs by sex, ontogenetic stage, general health, recent foraging success (and several other factors). Information regarding focal birds is needed to account for potential confounding factors.

L129-133: Some information on the number of trials that not included in the analysis is needed. Overall, how likely was it that gulls simply did not approach the food items, and was this confounded by the presence of competitors?

L121: how reliable is this 'mental notation'?

L134-141: the behavioural measures need to be properly (and completely defined). How long did it take for the gull to approach? Was this included as a dependent variable?

L155: as described in the methods, the objects were presented at a uniform distance. Why is this included as a factor in the model?

L 241: define JND

The discussion is often highly speculative. For example, what is the justification that the food items provided are novel (L276)? Presumably, gulls are pretty good at generalizing food preferences; how certain are you that they had not previously foraging on a similar food item? Likewise, lines 308-314 speculates on the role of individual differences... was this in anyway quantified? Generally, the discussion is overly long (5 pages) and could easily be shortened considerably to focus only on the relevant data.

Author's Response to Decision Letter for (RSOS-191959.R0)

See Appendix A.

RSOS-191959.R1 (Revision)

Review form: Reviewer 2

Is the manuscript scientifically sound in its present form?

Yes

Are the interpretations and conclusions justified by the results?

Yes

Is the language acceptable?

Yes

Do you have any ethical concerns with this paper?

No

Have you any concerns about statistical analyses in this paper?

No

Recommendation?

Accept as is

Comments to the Author(s)

Thank you for your work in revising this paper. I'm sure it will be of considerable interest to the journal readers!

Decision letter (RSOS-191959.R1)

03-Feb-2020

Dear Miss Goumas,

It is a pleasure to accept your manuscript entitled "Urban herring gulls use human behavioural cues to locate food" in its current form for publication in Royal Society Open Science. The comments of the reviewer(s) who reviewed your manuscript are included at the foot of this letter.

on behalf of Kevin Padian (Subject Editor)
openscience@royalsociety.org

Reviewer comments to Author:

Reviewer: 2

Comments to the Author(s)

Thank you for your work in revising this paper. I'm sure it will be of considerable interest to the journal readers!

Appendix A

Goumas, M., Boogert, N. J. and Kelley, L. A. response to reviewer comments

We thank the Editors for considering our manuscript “Urban herring gulls use human behavioural cues to locate food” for publication in *Royal Society Open Science*. We are also very grateful for the comments from the two reviewers, which we address below. Our responses are in bold italic below the reviewers’ comments. Our changes to the manuscript are highlighted in yellow.

Reviewer 1’s comments:

My sole suggestion is that the authors not rely so heavily of the p-value. I cannot deny that it has long been acceptable to report a small p-value, reject H_0 , and conclude H_A is supported, but it remains that what matters is not the probability of a type I error so much as how biologically meaningful is the result. The authors therefore ought to report an estimate of the effect size. This estimate is not trivial for a logistic regression model (I use this term because a GLM with binomial errors and a logistic regression are interchangeable in terms of maximum likelihood estimators generated), yet neither are they particularly difficult to calculate (see, for example, Allen & Le 2008, *J. Edu. Behav. Stat.* 33:416–441). Short of that, an effect size estimate for a basic observed vs. expected goodness-of-fit test, with expected assumed to be equal pecks at handled or not, is easy to obtain. The need for estimates of effect size are underscored by the authors own remark (p. 11) that “Although more gulls pecked at the handled food object compared to the handled non-food object, the total number of gulls pecking at either of the objects was similar in the food and non-food object trials.” If the effect size was roughly the same, then conclusions ought to be revised.

We thank the reviewer for this suggestion, and for providing a helpful reference to a paper explaining how to calculate effect sizes. We agree that reporting the effect sizes would improve the quality of the manuscript. We now include the odds ratios, calculated by taking the exponential of the regression estimates (please see lines 166-171 in the Methods, our reporting of results and Table 1).

I found results of the second experiment less compelling, perhaps because I wonder about the curiosity factor. I agree that a key difference is that the item presented in exp. 1 was food whereas in exp. 2 it was not, but another key difference, if I understand the experimental set up properly and judging from images in Fig. 2, is that the items in the first instances were wrapped whereas the item in the second instance was not. Would presentation of a wrapped non-food item have elicited comparable results to exp. 1? Similarly, would presentation of an unwrapped food item in exp. 1 have elicited different results? Could it be that urban gulls are naturally curious about a wrapped item—they can discern something is inside—and more likely to peck at a wrapped item another organism handled? In asking these questions I do not mean to suggest that results ought to be called into question but rather this possibility be discussed.

We agree that food packaging (wrapping) may affect gulls' responses and that an experiment disentangling the effect of a food source and food packaging (i.e. an object that may have strong associations with food) would provide a greater insight into the cues that gulls in urban areas use in their exploratory and foraging behaviour. The aim of Experiment 2 was to test the effect of handling an object that did not resemble food or something food-related, so that gulls' responses would not be based on prior observations of food or food packaging (if gulls categorise items in such a way). If the object had been reminiscent of food, this might have attracted the gulls for that reason alone rather than placing importance on the behavioural cues of a human handling the object. We now address the question of whether gulls are drawn just to food or can be drawn to food packaging itself in Discussion lines 285-287.

Lastly, and with all due respect for the hard work it entailed, I do not see how the visual analysis adds appreciably to the story. Put another way, if it were removed there would be little need to change much of the Results or Discussion.

We included the visual analysis to show that the gulls were capable of detecting both the food and non-food objects against the background, and they were able to discriminate between them, given their visual system. This analysis thus shows that the gulls' perception of the two types of objects is unlikely to have biased the results. However, we realise that this is supplemental to the rest of the data and, as such, have now placed this section in the Supplementary Materials.

Reviewer 2's comments:

Additional details are needed regarding the focal subjects. What was the local density of gulls for each observation? Social information theory predicts differential use of public information based on costs. Competitive costs would likely be much higher (leading to a greater use of social information) in aggregations vs. solitary or small groups of birds.

We did not record the local density of birds at each location, but there were no instances of large aggregations of birds. This is most likely because we targeted individual adult gulls during the breeding season when they defend discrete territories; aggregations tend to consist of non-breeders (mainly immature individuals) away from defended territories (pers. obs. and Tinbergen 1953). We were very conscious of avoiding interference from conspecifics and heterospecifics. We therefore did not conduct trials when there was likely to be such interference, and we ended trials that were interrupted by con- and heterospecifics. We cannot rule out that the presence of other gulls in the vicinity may have prevented certain gulls from approaching the experimental objects, and this may partly explain whether or not gulls approached. However, we would expect that the chance of this occurring would be approximately even in food and non-food trials, and is unlikely to have biased the results.

Likewise, social information use differs by sex, ontogenetic stage, general health, recent foraging success (and several other factors). Information regarding focal birds is needed to account for potential confounding factors.

As we are studying free-living wild animals, there are limitations in the number of confounding variables that we can account for. At present, we do not have a large population of ringed herring gulls in the region, and we have little information on the ringed gulls that we do have. We thus focused on sampling as many gulls as we could while ensuring that we did not sample the same individual twice and did not test neighbouring territory holders. This is likely to have minimised any potential effect of social information use in our experiments.

We targeted adult herring gulls (four years old or older) as evidenced by their adult plumage. Unfortunately, while we can be confident that we did not test immature gulls, we are not able to estimate the gulls' ages once their plumage is mature.

We recorded the sex of gulls where it was possible to do so, judging by size and behavioural differences. However, it was often not possible to do this, as it requires comparing pair members due to their small sexual dimorphism. We therefore did not include this information in our analyses due to the sample size being too small.

With regards to general health, we were not able to assess this except by visual inspection. None of the test subjects appeared to be in poor condition (e.g. poor feather quality, lack of alertness).

We were also not able to measure previous foraging success, and we assume that lack of hunger may in part explain why some gulls did not approach the objects. We cannot think of a reasonable hypothesis for why foraging history may affect whether gulls peck at handled versus non-handled objects, however, which is our main measure of interest.

L129-133: Some information on the number of trials that not included in the analysis is needed. Overall, how likely was it that gulls simply did not approach the food items, and was this confounded by the presence of competitors?

We report these details in the Supplementary Materials (under Results) and our analysis (line 199) shows how likely it was that gulls that remained in the area during the trials approached. There are probably many reasons for gulls not to approach during our trials, and we agree that the presence of competitors may be one reason. However, we were careful about targeting gulls for our experiments: we did not conduct trials within 10 m of another gull (excluding mates) for this reason, which is a sufficient distance if the trial takes place within a gull's territory, as territories are defended from conspecifics. We terminated trials if there was interference from other gulls, although this was relatively rare.

L121: how reliable is this 'mental notation'?

We used the video camera footage to confirm positioning of the gull. We now include this information in this line.

L134-141: the behavioural measures need to be properly (and completely defined). How long did it take for the gull to approach? Was this included as a dependent variable?

We collected data on the time taken for gulls to approach but did not originally include this variable as we felt it may not add to the results or their interpretation. We have now included these data (lines 123-124 and 195-198).

L155: as described in the methods, the objects were presented at a uniform distance. Why is this included as a factor in the model?

We attempted to make all distances uniform but it was not possible to do so, mainly because measuring distances would have deterred the gulls. We therefore measured the distances (distance between objects and distance between experimenter and gull) after the trials in case they had an effect on the gulls' behaviour (explained from line 127 onwards).

L 241: define JND

We have now moved the visual analyses sections to the Supplementary Materials in accordance with Reviewer 1's comments. "JND" is no longer referred to in the manuscript, and it is defined at its first use in the Supplementary Materials as "Just Noticeable Differences".

The discussion is often highly speculative. For example, what is the justification that the food items provided are novel (L276)? Presumably, gulls are pretty good at generalizing food preferences; how certain are you that they had not previously foraging on a similar food item? Likewise, lines 308-314 speculates on the role of individual differences... was this in anyway quantified? Generally, the discussion is overly long (5 pages) and could easily be shortened considerably to focus only on the relevant data.

We thank the reviewer for this comment, and we agree that it should be shortened. We have now removed some sections of the discussion, including the part about individual differences. We wanted to explore possibilities that could explain our findings as it may be beneficial for establishing future research, so we have placed some of it in the Supplementary Materials, and we hope that our discussion is now more focused. We have now clarified the line regarding novelty (now 247).